# Near-Infrared Emission of HgTe Nanoplatelets Tuned by Pb-Doping

**DOI:** 10.3390/nano12234198

**Published:** 2022-11-25

**Authors:** Anastasiia V. Sokolova, Ivan D. Skurlov, Anton A. Babaev, Peter S. Perfenov, Maksim A. Miropoltsev, Denis V. Danilov, Mikhail A. Baranov, Ilya E. Kolesnikov, Aleksandra V. Koroleva, Evgeniy V. Zhizhin, Aleksandr P. Litvin, Anatoly V. Fedorov, Sergei A. Cherevkov

**Affiliations:** 1PhysNano Department, ITMO University, Saint Petersburg 197101, Russia; 2Research Park, Saint Petersburg State University, Saint Petersburg 199034, Russia; 3Laboratory of Quantum Processes and Measurements, ITMO University, Saint Petersburg 197101, Russia

**Keywords:** nanoplatelets, near-infrared emission, cation exchange, photosensitivity

## Abstract

Doping the semiconductor nanocrystals is one of the most effective ways to obtain unique materials suitable for high-performance next-generation optoelectronic devices. In this study, we demonstrate a novel nanomaterial for the near-infrared spectral region. To do this, we developed a partial cation exchange reaction on the HgTe nanoplatelets, substituting Hg cations with Pb cations. Under the optimized reaction conditions and Pb precursor ratio, a photoluminescence band shifts to ~1100 nm with a quantum yield of 22%. Based on steady-state and transient optical spectroscopies, we suggest a model of photoexcitation relaxation in the HgTe:Pb nanoplatelets. We also demonstrate that the thin films of doped nanoplatelets possess superior electric properties compared to their pristine counterparts. These findings show that Pb-doped HgTe nanoplatelets are new perspective material for application in both light-emitting and light-detection devices operating in the near-infrared spectral region.

## 1. Introduction

Zero- and one-dimensional colloidal semiconductor nanocrystals, such as colloidal quantum dots and quantum rods, respectively, have been intensively studied for several decades. Their unique optical and electrical properties, combined with well-established synthesis protocols, have led to their increasing use in various technological areas, from biomedicine to optoelectronics [1,2]. Simultaneously, considerable attention has been paid to two-dimensional (2D) colloidal semiconductor nanocrystals, or nanoplatelets (NPLs), where the translational motion of the charge carriers is confined only in one dimension. At present, cadmium chalcogenide nanoplatelets, such as CdSe or CdTe, are among the most widely studied due to the possibilities they provide for the engineering of more complex structures and devices with further application in photonics or optoelectronics [3].

Nanoplatelets are well-regarded for their two-dimensional geometry and unique narrow-band photoluminescence (PL), which originates from the lowest excitonic state. For a specific material, the PL energy is primarily defined by the NPL thickness and can, therefore, be tuned via alterations in the synthetic parameters [4,5]. However, colloidal nanoplatelets of cadmium chalcogenides usually have a PL in the visible region and do not allow for the fundamental transitions to be shifted to the near-infrared (NIR) spectral range [6,7]. NIR-emitting nanoplatelets are of high interest for many practical applications, such as the fabrication of photodetectors, light-emitting diodes (LEDs), NIR lasers, and more [8,9]. Materials with a smaller band gap, e.g., lead chalcogenides, can be used to solve this issue, but their direct synthesis is more difficult and often lacks reproducibility [10].

More prominent synthetic routes are those where the IR-emitting nanoparticles are obtained from pre-existing host nanoparticles with visible range transitions via a complete or partial cation exchange [11]. Multiple publications describe the synthesis mechanisms and chemical substitution reaction kinetics in zero- and one-dimensional colloidal semiconductor nanostructures [12]. For instance, in a work by Knowles et al., the Cu^+^ ions were introduced into CdSe and InP quantum dots with a low degree of doping [13]. In either case, the dopant induces noticeable changes in the optical properties of the host material, including a shift in the emission band to the NIR region. Similar techniques may be applied to the 2D colloidal nanoplatelets. Galle et al. reported the complete cation exchange in CdSe NPLs to obtain PbSe nanoplatelets that are characterized by the broad redshifted emission band [8,14]. In a different work, Izquierdo et al. performed an analogous procedure on CdTe nanoplatelets with a mercury source, which resulted in the formation of HgTe NPLs with retained 2D geometry [15]. A reasonable continuation of this path is the doping of HgTe nanoplatelets with Pb that will further shift their photoluminescence to longer wavelengths. Such studies are very important for the emergence of new optoelectronic devices that operate in the near-infrared spectral window.

Herein, we report the three-step synthesis and characterization of the NIR-emitting lead-doped mercury telluride (HgTe:Pb) nanoplatelets. At the first step, three-monolayer-thick HgTe NPLs were prepared by a well-established cation exchange procedure from CdTe nanocrystals. At the second step, HgTe NPLs were treated by lead thiocyanate solution to induce the partial cation exchange of mercury with lead. This resulted in the formation of HgTe:Pb nanoplatelets with a preserved morphology but different optical properties that depend on the doping degree. The synthesized nanoplatelets were characterized by the steady-state and time-resolved emission spectroscopy. The measurements of charge carrier mobility and photoresistance of HgTe:Pb NPLs were also performed, which are relevant for the application of these nanoparticles in optoelectronics. We believe that our study will help to develop new nanostructure-based functional devices operating in the near-infrared spectral region.

## 2. Materials and Methods

The synthesis of colloidal nanoplatelets takes place in three steps, which are schematically shown in Figure 1. For the synthesis of CdTe NPLs and the cation exchange procedures to obtain HgTe NPLs, we used the methods from [5,15], respectively. Detailed syntesis protocols are presented in the Appendix A. A detailed list of used chemicals can also be found in SI.

### 2.1. Synthesis of HgTe:Pb Nanoplatelets

The technique for obtaining ternary NPLs is as follows. A mixture of 300 μL of oleic acid and 6 mL of 1-octadecene was degassed at 80 °C for 1 h. Simultaneously, the lead precursor was prepared by dissolution of lead thiocyanate in trioctylamine at a concentration of 8 mg/mL. Pb concentration in HgTe:x%Pb nanoplatelets varied according to the changing molar ratio x% of Pb:Hg. The reaction flask with mixture of oleic acid in 1-octadecene and Pb(SCN)_2_ in trioctylamine was additionally degassed for 10 min, after which it was transferred to an inert atmosphere. Then, 300 µL of the HgTe NPLs stock solution diluted to 2 mL with toluene was injected. After 30 min, the reaction was stopped by cooling with a water bath crude solution was centrifuged at 5000 rpm for 5 min. Doped NPLs were dissolved in tetrachlorethylene for further optical characterization.

### 2.2. Characterization

A Libra 200FE (Zeiss, Oberkochen, Germany) transmission electron microscope (TEM) was used to study NPLs morphology and dimensions. XPS measurements were performed on an Escalab 250Xi (Thermo Fisher Scientific, Waltham, MA, USA) photoelectron spectrometer with AlKα radiation (photon energy 1486.6 eV). Measurements were made in the constant pass energy mode at 100 eV for the survey XPS spectrum and at 50 eV for the core-level spectra of single elements, using an XPS spot size of 650 μm.

All spectroscopic studies were performed under an ambient atmosphere. For absorption and PL measurements, NPLs were dissolved in tetrachloroethylene, which is transparent in the NIR. Absorption was measured using a Shimadzu UV3600 spectrophotometer (Shimadzu, Kyoto, Japan). NIR PL spectra were measured using custom-built setup [16] with an excitation at 633 nm. PL spectra of CdTe and HgTe NPLs in the visible range were measured using 405 nm continuous wave laser (Lasever Inc, Ningbo, China) as an excitation source and Si-based CCD-camera (Andor iDus 401, Andor, Belfast, Northern Ireland) as a detector. All spectral data were corrected by the spectral sensitivity of the setup obtained by using the standard spectrum of an ideal black body [17]. Time-resolved photoluminescence (TRPL) measurements in the NIR were carried out using a single-photon avalanche diode (Micro Photon Devices, Bolzano, Italy) synchronized with 635 nm pulsed (~100 ps, 25 kHz) laser (PicoQuant, Berlin, Germany) as an excitation source [18].

For the charge carrier mobility investigation, the samples were prepared on a Prefabricated OFET Test Chips (Ossila, Sheffield, United Kingdom), the substrates based on highly doped Si-chip with prepatterned metallic electrodes. Channel width was 1 mm, channel length was 30 μm, silicon oxide dielectric layer thickness was 300 nm, specific capacitance is 1.09 × 10^−8^ F/cm^2^. For electron mobility investigation, silanization is required because the silicon oxide tends to form acceptor trap states. The silanization was performed using hexamethyldisilazane (HMDS). Substrates were soaked for 20 min in a diluted solution (0.1 mL of HDMS per 4 mL of cyclohexane) and rinsed by cyclohexane three times.

The HgTe and HgTe:Pb NPL films were spin-coated onto the substrates with a subsequent exchange from oleic acid to EDT (1,2-ethanedithiol, 0.2 vol% in acetonitrile). The NPLs solution was spin-coated at 2000 rpm (droplet volume 30 μL), after that EDT solution was casted onto the sample and spin-rinsed for 30 s with acetonitrile. NPLs layers of 80–100 nm were obtained as a result. Measurements were made with Keithley 2636b SourceMeter (Keithley Instruments, Cleveland, OH, USA).

For investigation of photoresistance of HgTe:Pb-EDT NPLs, nanoparticles were deposited on ITO glass substrates (Ossila, Sheffield, United Kingdom) with prepatterned electrodes were used for the study. Their channels have width of 30 mm and length of 50 μm. Keithley 2636B SourceMeter (Keithley Instruments, Cleveland, OH, USA) was employed for this research, the light source was a solar radiation simulator based on a xenon lamp with the AM1.5 filter and illumination intensity of 1000 W/m^2^.

## 3. Results and Discussion

### 3.1. The Influence of Pb Doping on the Morphology of HgTe Nanoplatelets

The synthesis of Pb-doped HgTe nanoplatelets described in this report consists of two cation exchange processes. The first step is the preparation of stable HgTe NPLs by a complete exchange of cadmium in the initial three-monolayer-thick CdTe NPLs with mercury. This was achieved via the reaction between CdTe nanoplatelets and a hard basic salt of mercury, as detailed in Reference [15]. According to the protocol in [15], either oleylamine or trioctylamine can be used to dissolve a hard mercury salt (in our case, Hg(Ac)_2_). We chose trioctylamine because it does not bind to the surface of NPLs, which enables precise control of the surface chemistry at each step of the cation exchange procedure. Moreover, because of the lower surface activity, the TOA-Hg(Ac)_2_ solution is more tolerant to possible deviations of the Hg/Cd ratio, making the procedure easier to perform without a noticeable loss of quality of the resulting product. In the end of the cation exchange procedure, oleic acid was added to reestablish the surface of NPLs. The choice of ligands for the as-prepared HgTe nanoplatelets plays an important role later, in the next cation exchange. Generally, oleic acid, oleylamine, or dodecanethiol can be used to stabilize HgTe NPLs [19,20]. Dodecanethiol is not suitable because its high binding energy to the NPL surface atoms might hamper the next cation exchange reaction [7]. Oleylamine was also disregarded, as it leads to a sharp increase in the instantaneous reaction region, resulting in a poor sample crystallinity [14].

Complete transition from CdTe to HgTe is reflected by a significant shift in both PL and absorption bands from visible to the near infrared. The absorption spectrum of pristine HgTe NPLs has two sharp maxima corresponding to the two excitonic transitions: electron-heavy hole at 820 nm and electron-light hole at 700 nm [21] (see Appendix A). The PL spectrum of HgTe NPLs is characterized by one symmetrically shaped narrow band positioned at 830 nm, a small Stokes shift of ~10 nm, and a full width at half maximum (FWHM) of ~35 nm. Therefore, although bands position is significantly red-shifted, the bands shape remained similar, which indicates the preservation of the two-dimensional geometry, as well as the good crystallinity of the sample. This is also supported by TEM measurements of CdTe and HgTe nanoplatelets (see insets on Appendix A). These data agree well with the previous reports [21].

The Pb-doped HgTe nanoplatelets (HgTe:Pb) were prepared at the second step via the partial exchange of Hg with Pb by lead thiocyanate dissolved in TOA. The amount of lead doping in NPLs was varied using different Pb/Hg molar ratios, from 1% to 6%. The latter was calculated from the chemical composition of the lead precursor and from the composition and the molar concentration of the initially prepared HgTe nanoplatelets. The determination of the Pb/Hg ratio is detailed in the Appendix A. For clarity, the samples will be named according to the percentage molar ratio of the Pb precursor (1−6%). Pb(SCN)_2_ was chosen as a lead source because it satisfies the principle of hard and soft acids and bases, which drives the reaction of cation exchange. Other compounds, such as lead iodide or lead bromide, have been considered as alternatives, but the treatment of NPLs with their solutions did not lead to any noticeable changes, even at high Pb/Hg ratios, indicating that they are not suitable to initiate the desired reaction. Notably, under ambient conditions the partial cation exchange with Pb(SCN)_2_ takes more than 12 h. It has been experimentally determined that by increasing the temperature to 75 °C, it is possible to reduce the reaction time to 30 min without any loss in the resulting parameters of the HgTe:Pb NPLs.

To avoid crystal lattice distortion in the host material, the Pb/Hg ratio during the cation exchange was kept under 10%. However, as seen from transmission electron microscopy images (Figure 2a,b and Appendix A), nanoplatelet morphology degradation occurs at the Pb/Hg ratio of 5% and higher. TEM images of HgTe:Pb NPLs with a lower percentage of doping (≤4%) demonstrate that these samples retain the shape and structure of the host mercury telluride NPLs.

All Pb-doped samples contain Hg, Te and Pb as seen from the full XPS survey (Appendix A). The atomic percentage of the samples is calculated from the XPS data, and summarized in Appendix A. Therefore, we can observe that NPLs were successfully doped with Pb, while the actual Hg-to-Pb substitution percentage varied from 0.7% to 1.2% depending on the sample. The high-resolution XPS spectra of the HgTe:Pb 4% sample are shown in Figure 2c. The sample demonstrates XPS peaks attributed to Hg 4f (104.2 ± 0.1 eV, 104.2 ± 0.1 eV), Te 3d (582.7 ± 0.1 eV, 572.3 ± 0.1 eV), and Pb 4f (142.7 ± 0.1 eV, 138.4 ± 0.1 eV). We also observe the peak in the SiO_2_ substrate (Si 2p 102.3 ± 0.1 eV) in the Hg 4f region. The estimated positions of the corresponding peaks agree well with the literature data [22,23].

### 3.2. Spectral and Temporal Emission Parameters of the HgTe:Pb Nanoplatelets

The HgTe:Pb NPLs were synthesized from HgTe NPLs via the partial cation exchange of mercury with lead. Even with a partial exchange, the optical properties of the final product may differ substantially from those of the host material due to the crystal lattice and energy structure modifications [12,24,25]. Cation exchange reaction dynamic was monitored by optical measurements of the aliquots that were sequentially collected over specific time periods. To do this, we separately performed an exchange reaction on HgTe NPLs with a Pb/Hg ratio of 4% and the temperature of 50 °C. The latter allowed for us to moderately slow down the reaction, and thereby simplify the aliquot collection.

The PL spectrum of the HgTe:Pb NPL exhibits noticeable changes, even after one minute of the reaction time. A new NIR band appeared in the emission spectrum of the NPLs (see Figure 3a). The intensity of the PL band at 830 nm (labeled PL1) expectedly decreases with the reaction time, whereas the intensity of the red-shifted PL (labeled PL2) band increases alongside its further shift toward the infrared (see Figure 3b), reflecting the gradual Pb atoms’ diffusion into the host material structure.

To monitor the emission kinetics, we recorded PL decay curves from the HgTe fundamental transition, and the intensity-averaged PL decay time ($\tau_{\mathrm{avg}}$) was calculated for each aliquot (see Appendix A for details). The PL decay time registered at 830 nm (Figure 3c) remains almost constant throughout the reaction. Since there is no such scenario where an additional relaxation pathway does not shorten PL lifetime, we speculate that any amount of Pb atoms totally quenches the HgTe NPLs emission at 830 nm. Therefore, we attributed the PL1 band to the undoped NPLs, and Figure 3b can also represent the ratio of undoped/doped NPLs throughout the reaction time.

For the HgTe:Pb NPLs samples with a Pb/Hg ratio from 1% to 6%, the emission spectrum of the NPLs significantly changes upon an increased amount of lead doping (see Figure 4a). NPLs with 1% and 2% doping demonstrate a well-resolved PL component centered at 830, indicating the remaining undoped NPLs. These are a quite conventional phenomena, and have been observed in similar works [26,27]. On the other hand, absorbance spectrum remains almost unchanged; the electron-heavy hole and the electron-light hole absorption bands retain their spectral positions for all HgTe:Pb NPL samples. However, the bands become broader at higher Pb/Hg ratios, indicating the reduction in the transition oscillator strength, which, in turn, can be a sign of a strong HgTe crystal lattice distortion due to the diffusion of a large amount of lead atoms. This observation is further supported by the TEM images presented in Appendix A, where the degradation of the nanoplatelet morphology is seen for the samples prepared with 5% and 6% lead doping.

The PL spectrum of the HgTe:Pb NPLs exhibits noticeable changes, starting from the Pb/Hg ratio of 1%. The lead-associated new NIR band in the NPLs emission spectrum (see Figure 4a) is centered at around ~1000 nm and is characterized by a large FWHM of ~150 nm and an effective Stokes shift of ~170 nm. The PL QY of the NIR emission was determined to be 7% (Figure 4b). A further increase in the Pb/Hg ratio up to 4% leads to a red shift of this NIR PL band, accompanied by the PL QY increase of up to 22%. At the same time, the primary HgTe PL band positioned at 830 nm rapidly decreases and eventually disappears at the Pb/Hg ratio of 2%. When the Pb/Hg ratio exceeds 4%, the quantum yield of the NIR PL band of the HgTe:Pb nanoplatelets starts to decrease, indicating their degradation, as previously discussed.

To analyze the effect of Pb doping on the charge carrier recombination kinetics, the PL decay curves were registered for all HgTe:Pb NPL samples at the spectral region of the NIR emission band (typical PL decay curves can be found in Appendix A). The PL kinetics of the HgTe:Pb nanoplatelets is multiexponential, indicating the possible involvement of the intra-band gap energy states that originate from lattice imperfections or surface defects.

Averaged PL decay time (*τ_avg_*) change (Figure 4c) resembles the change in the PL QY: *τ_avg_* increases when the Pb/Hg ratio rises from 1% to 4%; however, it rapidly decreases when the Pb/Hg ratio exceeds 4%. The PL lifetime and QY are related via a well-known expression:$$PL\;QY=\frac{\tau_{avg}}{\tau_{i}},$$
where $\tau_{avg}$ is the measured PL lifetime and $\tau_{i}$ is an intrinsic PL lifetime [28]. We believe that the disproportion between PL QY and $\tau_{avg}$ is caused by two factors. Firstly, a change in the wavelength-dependent intrinsic recombination rates (i.e., $\tau_{i}$). Secondly, for 1% and 2% HgTe:Pb samples, there are emissions from HgTe excitonic state that are not accounted for in QY but increase the absorption.

Based on the obtained spectroscopic results, we propose the HgTe:Pb NPLs energy relaxation model, schematized in Figure 5. The initial HgTe NPLs possess the conventional PL from the excitonic transition. The HgTe band structure suffers no significant changes during Pb doping, which clearly follows from the absorption spectra. Pb doping introduces the intraband states that swiftly trap one of the charge carriers, quenching the PL of the excitonic transition. The observed NIR PL arises from the recombination of holes/electrons from the main state with the holes/electrons from the Pb-related state. The actual type of charge carriers is irrelevant for the qualitative description. An increase in the amount of dopant leads to a reduction in the energy of the corresponding radiative transition. A similar dependance was observed for Cd-based NPLs doping with Cu and Ag atoms [26,27]. In case of Cu, the redshift is caused by a shift in the conduction band edge via orbital hybridization with dopant atoms. However, this complex theoretical investigation of HgTe:Pb interaction is beyond this paper’s scope.

### 3.3. Electronic Properties of HgTe:Pb Nanoplatelets

For optoelectronics applications, e.g., the fabrication of photodetectors or LEDs, it is important to obtain information about the charge carrier mobility in the synthesized Pb-doped HgTe nanoplatelets. In this section, we used the field-effect transistor method to determine the charge carrier mobility in the studied HgTe:Pb NPLs layer. In this method, the studied material forms a channel between the drain and the source of the transistor. The gate is separated from the channel with a layer of dielectric material, and the electric potential of the gate either repels or attracts charge carriers in the channel. This interaction affects the charge carrier concentration, consequently changing the channel resistance and the drain-source current [29]. The hole mobility can be determined from the p-channel mode with a negative bias for both gate and drain, vice versa the electron mobility is determined from the n-channel mode with a positive gate and drain bias.

For pristine HgTe sample p-channel mode, we observe an increase in the current with the *V_G_* growth, which indicates the presence of hole conductivity (see Figure 6a). In n-channel mode the current does not change with the *V_G_* growth, implying that there is no gate influence on conductivity. This might indicate the absence of electron conductivity as well as the channel-insulator interface polarization [30]. A significant change in conductivity occurs at high gate voltages (*V_G_*) and drain source voltages (*V_DS_*). The output characteristics of other samples did not significantly differ from this one. Output characteristics *I_DS_*(*V_DS_*) of the reference HgTe-EDT sample with different *V_G_* are linear in the region of 0 V. This means there is no influence from contact resistance, and we can use Equation (1), as below.

By analyzing the dependence of the drain-source current (*I_DS_*) on the gate potential (*V_G_*), it is possible to estimate the mobility of electrons and holes. Charge carrier mobility can be defined by Equation (1):(1)$$\mu =\frac{L}{WC_{ox}V_{DS}}\frac{\partial I_{DS}}{\partial V_{G}}{|}_{V_{DS}=const}$$
where *W* is the channel width, *L* is the channel length, *C_ox_* is the specific capacitance of the insulator layer, *µ* is the carrier mobility, (*∂I_DS_/∂V_G_*) is the linear fit slope of the transfer characteristic *I_DS_*(*V_G_*). The samples’ thickness is assumed to be roughly equal. Mobility measurements were averaged by the five contact points on the sample.

P-channel mode transfer characteristics of HgTe:Pb-EDT NPLs layer were obtained with high *V_DS_* = 80 V, which indicates poor layer conductivity. The polarization effects were prevented by utilizing the relatively high (~5 V/s) *V_G_* change speed. The gate current *I_G_* is about 10 times lower than the drain-source current *I_DS_*, so gate polarization can also be neglected. All the samples show monotonous linear |*I_DS_*| growth, so it is possible to estimate the hole mobility from linear fitting. Transfer curves for HgTe:Pb (0%, 2% and 4% Pb) NPLs and their linear fitting are demonstrated in Figure 6b. The n-channel transfer curves of a HgTe-EDT sample (Appendix A) show a monotonous decrease, which indicates the absence of electronic conductivity in this voltage range.

As the doping level grows, the transfer curve incline increases in the hole conductivity region (see Figure 6b). Furthermore, in the case of 4% Pb doping, the overall conductivity of the sample significantly grows. The calculated conductivity was 7.0 × 10^−7^, 1.5 × 10^−6^ and 3.8 × 10^−6^ cm^2^/(V·s) for the samples with 0%, 2%, and 4% doping levels, respectively. Thus, we may speculate that HgTe nanoplatelets Pb doping leads to an increase in a hole mobility, which, in turn, leads to the growth in the potential efficiency of photodetectors based on these NPLs.

HgTe nanoplatelets can be used as photoresistors in light detectors [21,31]. An increase in charge carrier mobility with Pb doping should also impact the photosensitivity of the studied NPLs. Hence, we performed IV-measurements of pristine and Pb doped HgTe NPLs to determine their photoconductivity.

To obtain the maximal light absorption, the samples were fabricated with a drop-casting method, resulting film thickness was 300–400 nm (see Table 1 for the values). IV-characteristics were obtained in the dark and under the AM1.5 solar simulator illumination, and were averaged from five contacts on the sample. The IV curves are demonstrated in Figure 5c,d. The averaged current values in the dark (*I_D_*) and under illumination (*I_L_*) at 20 V are summarized in Table 1.

We used two values to compare samples with different doping levels: photosensitivity, determined as *S* = (*I_L_* − *I_D_*)/*I_L_* [32], and relative conductivity change (*R_L_*/*R_D_*). The dark current increases together with the doping level, which is another sign of the charge (hole) carrier mobility growth. Doping does not significantly improve sensitivity; however, it can be useful for dynamic measurements due to its higher conductivity and shorter rise time.

## 4. Conclusions

To conclude, we demonstrated that a partial cation exchange in HgTe nanoplatelets with Pb atoms can achieve a new efficient near-infrared emitting nanomaterial. A careful choice of doping ratio allows for modification of the optical and electrical properties of the nanoplatelets without damaging their morphology. As a result of gradual doping, lead atoms form intraband electronic states, leading to a quenching of the band-edge emission and enhancement of the dopant-related photoluminescence in the near-infrared spectral region. Thin films of HgTe:Pb nanoplatelets possess enhanced hole mobility and conductivity, both increasing with a Pb doping level. The reported doping strategy makes Pb-doped HgTe nanoplatelets a promising material for the development of next-generation near-infrared photodetectors, LEDs, and other optoelectronic devices.

## Figures and Tables

**Figure 1 nanomaterials-12-04198-f001:**
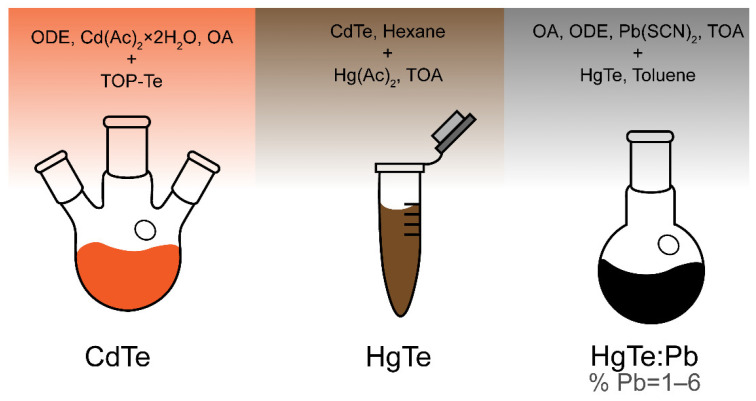
Step-by-step scheme of colloidal HgTe:Pb synthesis.

**Figure 2 nanomaterials-12-04198-f002:**
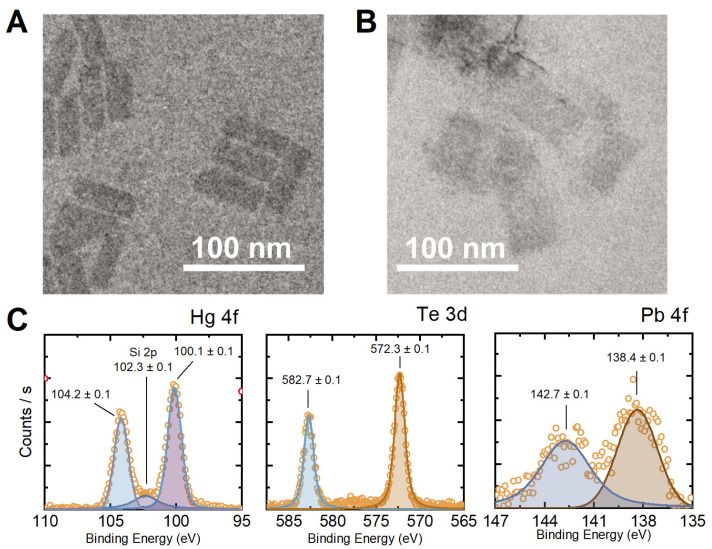
(**A**,**B**)—TEM images of the HgTe and 4% Pb-dopped HgTe NPLs, respectively. (**C**)—High-resolution XPS spectra taken from HgTe:Pb 4% for Hg, Te and Pb elements.

**Figure 3 nanomaterials-12-04198-f003:**
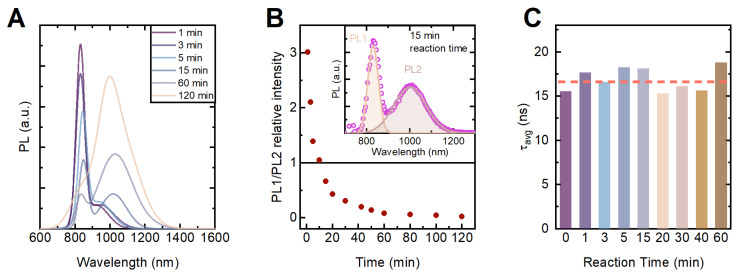
(**A**) Photoluminescence spectra; (**B**) relative PL intensity of 830 nm band (PL1) and >950 nm PL band (PL2), on inset–typical PL deconvolution on a 15 min-aliquot; (**C**) intensity-averaged PL decay times (at 830 nm) of 4% doped HgTe at different reaction time.

**Figure 4 nanomaterials-12-04198-f004:**
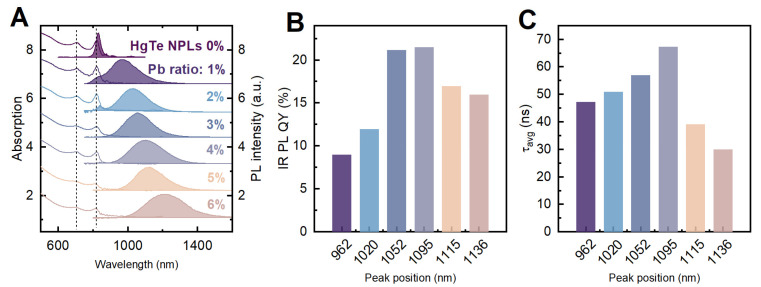
Optical properties of HgTe:Pb nanoplatelets with different lead ratios: (**A**) photoluminescence and absorption spectra; (**B**) near-infrared band PL QY vs. its peak position; (**C**) averaged near-infrared photoluminescence decay times vs. peak positions.

**Figure 5 nanomaterials-12-04198-f005:**
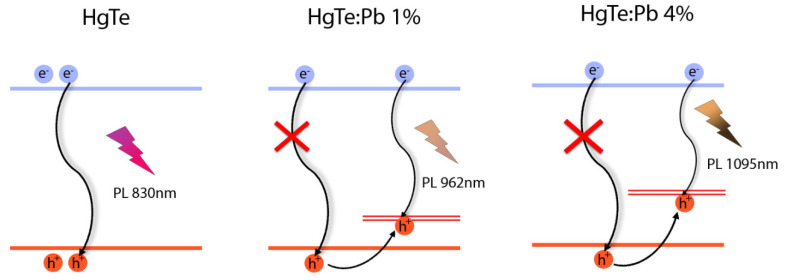
The scheme of the energy relaxation in the HgTe:Pb NPLs.

**Figure 6 nanomaterials-12-04198-f006:**
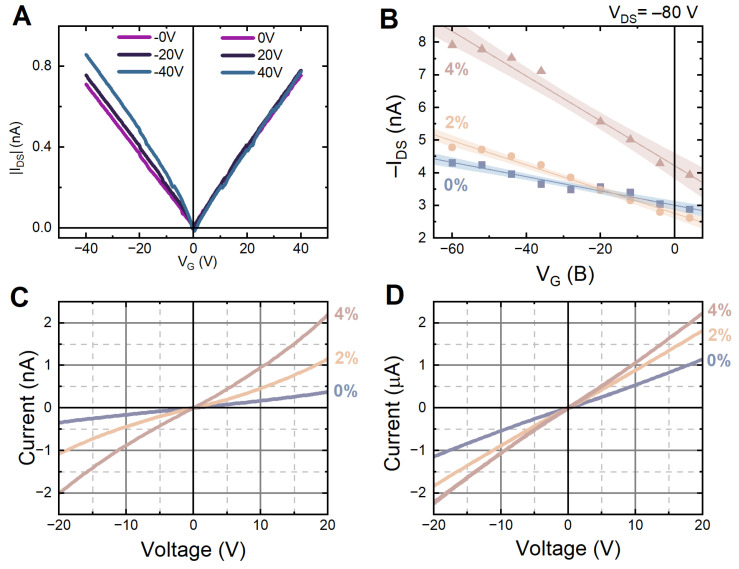
(**A**) Output characteristics *I_DS_*(*V_DS_*) of pristine HgTe-EDT with different *V_G_* values, (**B**) Averaged transfer curves of Pb-doped HgTe-EDT NPLs; IV curves of Pb-doped HgTe-EDT NPLs in the dark (**C**) and in the light (**D**).

**Table 1 nanomaterials-12-04198-t001:** Photoelectric parameters of HgTe-EDT:Pb NPLs layers.

Pb/Hg Ratio	Thickness, nm	I_D_, A	I_L_, A	Photosensitivity	R_L_/R_D_	Resistivity p_D_, Om·m	μ_h_, cm^2^/(V·s)
0%	340	3.6 × 10^−10^	1.1 × 10^−6^	0.9997	3060	11.3 × 10^6^	7.0 × 10^−7^
2%	590	1.2 × 10^−9^	1.8 × 10^−6^	0.9993	1500	5.9 × 10^6^	1.5 × 10^−6^
4%	435	2.2 × 10^−9^	2.2 × 10^−6^	0.9990	1000	2.4 × 10^6^	3.8 × 10^−6^

## Data Availability

Not applicable.

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
