# Peer review of "Near-Infrared Emission of HgTe Nanoplatelets Tuned by Pb-Doping"

_nanomaterials, 2022, doi:10.3390/nano12234198_

Round 1

Author Response

Dear Reviewer,

Reviewer 2 Report

The authors have presented a comprehensive study on the Pb-doped HgTe nanoplatelets including synthesis and characterization. Discussions are well organized and supported by the experimental data. Some minor revisions should be noted before acceptance for publication.

1.      The figure 1 presented a schematic diagram of synthesis. The doping levels changed from 1 to 6 %, not 1/6 %. Please note the symbol.

2.      The figure 5 presented a schematic diagram of photoluminescence. The notations of HgTe, HgTe 1% and HgTe 4% could be modified to indicate the Pb-doping levels, such as HgTe:Pb 1% and HgTe:Pb 4%.

3.      In equation 1, the subscript (lin) of carrier mobility has not been addressed. Maybe the subscript is not necessary here.

4.      The definition of photoconductive sensitivity S is given by (IL-ID)/IL, it would be not clear to see the enhancement by illumination. I suggest that authors can modify the definition of S and given by (IL-ID)/ID.

5.      In table 1, the resistivity PD has not been described clearly. Please improve it.

6.      In table 1, the value of hole mobility is very low (such as 7X10-7), please check it and make sure the measurements and calculation are all right. For example, using a standard device to check the whole procedure.

Author Response

Dear Reviewer, 
